# The Third Half: A Pilot Study Using Evidence-Based Psychological Strategies to Promote Well-Being among Doctoral Students

**DOI:** 10.3390/ijerph192416905

**Published:** 2022-12-16

**Authors:** Anna Muro, Iván Bonilla, Claudia Tejada-Gallardo, María Paola Jiménez-Villamizar, Ramon Cladellas, Antoni Sanz, Miquel Torregrossa

**Affiliations:** 1Department of Basic, Developmental and Educational Psychology, Autonomous University of Barcelona, 08193 Bellaterra, Spain; 2Serra Hunter Programme, Generalitat de Catalunya, 08017 Barcelona, Spain; 3Sports Research Institute, Autonomous University of Barcelona, 08193 Bellaterra, Spain; 4Department of Psychology, University of Lleida, 25001 Lleida, Spain; 5Faculty of Health Sciences, Universidad del Magdalena, Santa Marta 470001, Colombia

**Keywords:** well-being promotion, distress, doctoral students, mental health, research career

## Abstract

Over the last few years, various studies have reported decreasing well-being levels among doctoral students, who show a higher risk of suffering from psychological distress than the general population. Accordingly, European policies in higher education encourage well-being promotion programs among doctoral studies to enhance young researchers’ well-being. However, programs using evidence-based practices for well-being promotion are not yet generalised in public universities. The present study describes a pilot experience designed to evaluate the efficacy of a well-being program among doctoral candidates of a public European university, the Autonomous University of Barcelona. 25 doctoral students (67% women) participated in a pre-post study consisting of six sessions of 3 h each and structured by the big five criteria coming from evidence-based practices for well-being promotion: outdoor green spaces exposure, physical activity, gamification, mentoring, positive and coaching psychology techniques. Results showed how participants experienced significant increases in several indicators of emotional well-being and decreased psychological distress after the Third Half program. These positive pilot results encourage further research and future replications to assess the impact of this evidence-based psychological program among the academic community. Results also lead the way towards the creation of healthier academic workplaces by implementing cost-effective interventions that improve researchers’ psychosocial support and their overall well-being.

## 1. Introduction

Over the last few years, various reports have shown evidence of the worsening well-being and mental health among doctoral students and Early Career Researchers (ECR) working in the European Research Area (ERA) [1,2]. According to the previous literature, young researchers are very satisfied and motivated by their learning process. However, they also show that before the COVID-19 pandemic, between 32% and 42% were at high risk of developing mental health problems, with depression, anxiety, burnout, and cognitive exhaustion being the most frequent disorders among the youngest academic community members. Hence, a higher risk of suffering from mental health complications is present among the ECR [3,4,5]. Accordingly, different European strategies have emerged during the last few years with the objective of promoting the well-being of doctoral students [1,2].

The estimated incidence of mental illness in doctoral students of the ERA before the pandemic was 2.84 times higher than that observed in the generally highly educated adult population [5]. This trend was also reported in public Catalan universities, showing that researchers with temporary employment contracts reported worse mental health: despite being the youngest group, they showed more stress symptoms and less job satisfaction when compared to both professors and full-time researchers with stable contracts [6]. This situation has worsened because of the COVID-19 pandemic’s impact on mental health, showing that around 45% of the academic community presents anxious-depressive symptoms, a prevalence nine points higher than in the general population [7,8,9,10], and with students showing a prevalence that reaches 53% in Ibero-american countries, with loneliness being one of the main risk factors for psychological distress during the pandemic [11]. The situation is similar in doctoral students of other European countries such as Belgium [5] or the Netherlands [12], in which no less than 47% of PhD students were at risk of developing a psychiatric disorder and 39% showed severe symptoms of burnout. This tendency, also found in other universities of the world [3,13] calls for research staff, higher education institutions, funding bodies, stakeholders, and governments to work together and invest in structural actions towards the creation of healthier academic careers and academic careers that promote mental health, not only that of doctoral students, but of the entire academic community [2,3,6,12,13,14,15].

It should be noted that a significant proportion of research outcomes and universities’ excellence levels depend on the work and contributions of ECR, who often represent more than the 50% of university teams [16,17,18] and play a key role in economic growth, innovation, and the advancement of knowledge in the European Union [19]. However, their technical specialisation and their talent development mostly depend on competitively funded projects, which generates a high professional uncertainty. Despite this specialisation, low salaries, contractual uncertainty, relationships with the supervisor, workload and complexity, job insecurity, pressure to publish, the lack of institutional support or funding to lead projects are considered as working conditions and psychosocial factors that underlie the decreasing well-being among ECR [1,2,3,4,5,6,7,20]. This situation is accompanied by a significant academic dropout, as a recent study in Spanish universities reports: one-third of the active doctoral students withdrew from their doctoral training [21], while the prevalence of dropouts at an international level ranges from 50% to 70% [22,23]. The main reasons for this high dropout rate points to the difficulty of reconciling doctoral studies with personal and professional life (25%), social isolation (20%), demotivation (19%) or lack of institutional support (40%) [20,21]. Therefore, the implementation and assessment of top-down programs that promote more motivating, sustainable, and healthy work environments in doctoral studies should be a strategic priority for the ERA universities, as it is a long-sought demand of the entire community of doctoral students [1,2,24,25,26].

### Well-Being Promotion among Doctoral Students

Well-being is closely related to mental health, as conceived by the World Health Organization (WHO) [27], which defines this construct as an integral and essential component of health, with health being a state of complete physical, mental, and social well-being and not merely the absence of disease. Mental health is also defined as a state of well-being in which an individual realizes his or her own abilities, can cope with the normal stresses and emotions of life, can work productively and is able to contribute to his or her community. On this basis, the promotion, protection, and restoration of well-being and mental health can be regarded as a vital concern of communities and societies throughout the world. Well-being promotion is also one of the Sustainable Development Goals (SDG) for the 2030 Agenda [28], hence, it should be in the spotlight of all worldwide public health systems and a priority of educational policies that should work to upskill individuals in well-being management towards the prevention of psychological distress among the youngest populations, those who have been impacted the most by the COVID-19 pandemic [12]. In this regard, well-being programs in higher education should offer cost-benefit and evidence-based psychological strategies that (a) facilitate individuals’ well-being by increasing psychosocial support in educational settings, and (b) reduce their increased risk for suffering distress and prevent mental health problems. 

In this context, doctoral schools and universities are starting to implement well-being programs that complement technical training [29,30,31,32], but these programs are not yet generalised since it is first necessary to create synergies of coordination between administrative teams and doctoral studies staff at a local level. These actions should necessarily be focused on structuring and guaranteeing more psychosocial services and training to upskill students and employees in well-being and mental health management, and ultimately, to foster the organisational culture and working climates where young researchers develop [4,5,21,25,26,27]. Therefore, more institutional investment is needed to implement sustainable services and evidence-based training that upskills ECR in well-being and mental health management at their workplace. These training services are not yet widespread in the ERA, in part because their implementation depends mostly on grants and projects [1,2,25]. Furthermore, although there is a growing number of descriptive studies and some reviews on the current situation of ECRs’ mental health [3,32], a gap still exists in the implementation and quality assessment of institutional interventions targeted at increasing researchers’ well-being and mental health management; published studies are generally qualitative and only use some strategies such as online support [29], mentoring or coaching [2,30,31]. None of the published investigations include a multicomponent approach using other empirically validated techniques that have been demonstrated to have a significant impact not only in mental health prevention, but also in the promotion of well-being. Hence, the implementation and assessment of multicomponent programs that guarantee mental health education and well-being promotion is a systemic challenge for the entire research community that could use knowledge transfer and data evidence from the field of the psychology of well-being to design, implement, and assess those most suitable and effective for training among their research staff. 

In the context of implementing well-being policies at a local level among universities, the Autonomous University of Barcelona (UAB) started working towards a healthier and more sustainable campus in 2018 [33]. The UAB is one of the main public universities in Catalonia and Spain, with more than 40.000 students, of whom 4629 are doctoral students, and is recognized with the distinction of “Human Resources Excellence in Research” [34,35]. A specific Social Responsibility Unit works to promote improvements towards a healthier campus in collaboration with the Catalan Net of Healthy Universities [33], and has implemented several actions that, in collaboration with other units such as the Research Network in Mental Health [36], have facilitated the promotion of mental health and emotional well-being among the academic community. During the last few years, following the guidelines of the European Charter for Researchers [37], the UAB has been also working on the implementation of services aimed at improving the well-being and optimal development of researchers, a priority in ERA policies that also follows the accomplishment of the SDG set by the United Nations and by the WHO in educational and academic settings [27,38].

In this policy framework, we aimed to implement and assess a pilot experience of well-being promotion and mental health prevention using a multicomponent approach of evidence-based psychological strategies among doctoral studies, an approach referred to as the Third Half. Given the empirical evidence behind the design of the Third Half (that will be explained in detail in the methods section), we hypothesized that the Third Half would significantly increase ECRs’ well-being and decrease their psychological distress levels.

## 2. Materials and Methods

### 2.1. Program Design: The Third Half

In response to the psychological impact of the pandemic [9,11], during October 2021, the Doctoral School and Vice-Rectorate of Campus, Sustainability, and Territory of the UAB contacted the Mental Health Research Network [36] and requested the design and implementation of a well-being program among doctoral students, to be assessed as a pilot study during the second semester of the 2021–2022 academic year. This Network contacted the Unit of Coaching and Academic Support (UCAA-SPL) [39] that accepted the request, not just to design and implement a training program but also to assess its impact on doctoral students’ well-being. The UCAA is composed by PhD Psychologists and part of their members are also co-authors of the present study. They designed a program called the Third Half. Originally, the Third Half refers to a rugby and sports tradition that takes place after the match and brings together all the players of the two teams, who take the opportunity to offer themselves drinks and food and exchange opinions and considerations as happens between friends [40]. Rugby’s values and codes of conduct praise respect, solidarity, teamwork, unity, cordiality and friendship, and clean and fair play as core values of this sport. With the same purpose, the UCAA adapted this practice for the design of the intervention by hosting monthly meetings between ECR. The characteristics of the Third Half program are presented below:

*Frequency and duration*: 6 sessions of 3 h each (one per month, from February to July).

*Structure*: The first 2 h were destined for activities aimed at promoting well-being in outdoor spaces on the Campus and the third hour (more informal) aimed to facilitate social connection and peer support by having a drink or a snack in one of the bars or cafes of the UAB.

*Trainers*: All of them were PhD psychologists specialised in motivation and academic well-being. For the pilot study, there was a specialist in Positive Psychology and Coaching applied in educational environments with the collaboration of the Doctoral program in Health and Sport. A doctoral student and psychologist specialised in gamification was selected to support both the design of the activity and the implementation of the program’s activities.

*Goals*: The main goals were to increase well-being and improve psychosocial support to reduce occupational risks associated with the doctoral career. We aimed to facilitate a space where peers can have time to talk, reflect, and create an informal environment in which to share feelings, doubts, and some fun, too.

*Activity Design Criteria*: The design of each activity followed the criteria of the WHO [27] and the ReMO [1] regarding the need for implementing mental health interventions and prevention programs in higher education and research settings. All the activities were framed within the motivational model of self-determination [41] and the humanistic principles of education, where unconditional acceptance or non-judgment are key principles for learning processes [42,43,44,45,46]. Activities were designed following five criteria from evidence-based psychological interventions:**Gamified activities** [47,48,49]. Gamification is a strategy that applies the game principles. It is defined as the integration of game elements into non-game activities using their mechanics and aesthetics. The objective is to engage people, motivate action, and promote learning and problem solving while having fun [50]. It has been shown to be effective in educational and learning processes [51] where the use of game mechanics improves motivation and learning in formal and informal settings.**Outdoor activities in green spaces** [52,53]. Outdoor learning is defined as “that which lies beyond the walls of the interior” and has been shown to provide more meaningful, deep, and stimulating learning experiences that facilitate interest and motivation to learn. It is often considered that outdoor learning can provide opportunities in many subjects and support students’ personal, social, and emotional development. On the other hand, it has also been demonstrated that exposure to green and natural environments immediately facilitates relaxation and emotional well-being, and therefore promotes the comprehensive health of individuals. The allocations alternated between outdoor and green spaces on the Campus.**Positive Psychology and Coaching applied in Educational Settings** [54,55,56,57,58] The activity was framed within the framework of Positive Psychology applied in educational environments for the prevention and promotion of psychological well-being. We used exercises specific to interventions based on Positive Psychology [59] targeted at students’ population in educational settings. Techniques widely used in Coaching Psychology and specific to Education were also used, such as Socratic maieutic or the art of asking questions and conversations geared towards students’ development, setting goals and values, identifying personal strengths, or enhancing communication skills to build healthier relationships.**Physical activity** [59,60]. The practice of moderate physical activity is a protective factor for mental health at all stages of development and especially for students. A lack of physical activity worsens health, both physical and mental, and is considered a key cost-benefit strategy in well-being policies. Although the goal was not to make a physical activity training program, typical games that encourage behavioural activation, teamwork, and cooperation were proposed, such as: passing the ball, handkerchief game, relays, the blind guide, etc.**Peer-mentoring and peer support** [61,62]. In conventional mentoring, the student is matched with someone more senior in the organization or who has more experience in a particular area of interest. There is often an expectation of professional development. In peer mentoring, the mentor is usually someone with a similar background, just a little more advanced academically, and who can bring an alternative perspective to the career path. The additional social support that allows the student to share their worries, concerns or conflicts, facilitates their performance and emotional well-being, while fostering bonds of friendship between the participants. The mentor offers space and time for reflection and dialogue, listens, and guides. Collective mentoring also facilitates peer connection and diminishes feelings of social isolation, promoting a shared learning experience that simultaneously facilitates identification with others, friendship, altruism, and cooperation.

A more detailed description of the program can be seen at the Third Half: Design and Implementation booklet [63,64].

### 2.2. Procedure

On 25 January 2022, a call was made for participating in the Third Half, announcing the activity on the UAB website and sending an email with a Google Forms link to all doctoral students of the UAB Campus through the mailing list of the Doctorate School. The applications doubled the available number of participants (*n* = 40). Eligible participants (see below the eligibility criteria) answered a battery of questionnaires designed to evaluate their levels of well-being and psychological distress before starting the program and right after. It was possible to compare at the pre-test the results from the Third Half group with a control group that did not take part in the program because a pre-survey was administered cross-sectionally to all members of the doctoral community. This was done to observe if participants who were willing to participate were representative and comparable to the non-participants. To identify the possible presence of chronic or serious pathologies, and thus be able to adapt the activities in case of having doctoral students with functional diversity or other significant disorders that affected mobility, an item was included in the survey. Participation was voluntary and anonymous, and participants were informed about the use of the data with solely scientific and research purposes. The study was approved by the Doctoral School, by the Campus-SIS Unit of the Vice-Rectorate and by the Ethical Committee of the UAB under the code: CEEAH6007.

### 2.3. Participants

After sending more detailed information and once confirmation of participation was received, only 25 doctoral students could meet the eligibility criteria, which was being able to engage and assist, at least 75% of the programmed sessions and being part of a research team of the UAB, independent of their contractual situation. To note, none of the participants reported illnesses or significant disorders that affected their health and mobility, so none of the activities had to be adapted during the program. In any case, alternative activities were contemplated for the participation of doctoral students with functional or mobility problems who could not do some of the scheduled activities. It should be noted that the activities were voluntary and there was no obligation to participate. Also, participants could withdraw from the program at any time. Finally, only participants who answered all the questionnaires (72%) before and after their participation were included in the analyses.

The average age of the participants was 31 years (*SD* = 6.90; range = [25–60]), 67% were women, and 44% were international doctoral students. Table 1 shows the distribution of participants field of knowledge.

### 2.4. Instruments

A short survey was designed in collaboration with the UCAA team and administered to the doctoral students to explore their socio-demographic characteristics. Two items were included in order to be aware of the health status of the doctoral students, asking if they suffered from any ongoing chronic or acute illness that could affect their participation in the activities, especially those that included physical activity.

Also, the survey had six self-reported measures of well-being and psychological distress that are often used in research: 

**Brief Scale of Emotional Profiles** (Profile of Mood States—POMS) [64]. The POMS measures six moods based on 30 items: anger, fatigue, vigour, friendship, tension, and depression and has a Likert-type response from 0 to 5. These six moods are emotional states and, therefore, are variable and reactive to situations and context. Although they can be indicators of the presence of possible psychopathologies, they have no clinical relevance and only indicate emotional profiles at the time of measurement. The internal consistency of the POMS was Cronbach’s *α* = 0.91. 

**Positive and Negative Affect Scale** (PANAS) [65]. The PANAS measures a pattern of experiencing positive emotions relating to emotional well-being and the experience of negative emotions relating to emotional discomfort. It includes two subscales (i.e., positive and negative) assessed using a total of 20 items, 10 items each, and has a Likert-type response from 1 to 5. The internal consistency of the PANAS was Cronbach’s *α* = 0.86.

**State and Trait Anxiety Inventory** (STAI) [66]. The STAI has two subscales: state anxiety (at the moment) and trait anxiety (global personality). It contains a total of 40 items, 20 on each scale, and has Likert-type responses from 0 to 3. High scores alert to altered states related to anxiety, and low scores indicate emotional stability and the absence of anxiety. In the present study, we only administered the state anxiety subscale. The internal consistency of this scale was Cronbach’s *α* = 0.80.

**General Anxiety Disorder-2** (GAD-2) [67]. The GAD-2 briefly measures the presence of symptoms associated with generalized anxiety disorder. The scale consists of 2 items on a Likert-type response scale from 0 to 3 and is used to assess the presence of symptoms in the previous two weeks. The final score is calculated by adding the scores of the 2 items. This can range from 0 to 6 and can be used to assign a provisional diagnosis: no anxiety disorder (0–2) and probable anxiety disorder (3–6). The internal consistency of this scale was Cronbach’s *α* = 0.82.

**Patient Health Questionnaire-9** (PHQ-9) [68]. The PHQ-9 items follow the nine criteria specified in the DSM-IV diagnostic manual for screening for depressive disorder. It includes 9 items in a Likert-type response scale from 0 to 3 and is used to assess the presence of symptoms in the previous two weeks. The total scale score is 27, and scores of 10–14 points, 15–19 points, and 20–27 points indicate, respectively, moderate, moderately severe, and severe levels of depressive symptoms. The internal consistency of this scale was Cronbach’s *α* = 0.85.

### 2.5. Statistical Analyses and Quality Assessment

All analyses were calculated with the statistical program SPSS v.26. The means (and standard deviations) were calculated for each indicator of well-being and distress. The frequencies (percentages) of participants’ sociodemographic data were also calculated. Anxiety (GAD-2) and depression (PHQ-9) were dichotomized using the cut-off points aforementioned to calculate the prevalence of symptomatology in participants and non-participants. Between groups, analyses of variance (ANOVA) were first performed to check if there were baseline differences in the program participants compared to non-participants based on gender, age, family situation, field of knowledge, and cultural background (international/national) to identify possible confounders. T-tests were also performed to analyse if there were baseline differences among participants of the Third Half according to gender, cultural background, or field of knowledge. Next, the non-parametric Wilcoxon repeated-measures test was used, due to the small sample size, to analyse pre-post changes before and after the activity and thus explore the impact of the intervention on the different indicators of well-being and psychological distress of the Third Half program participants. Additionally, an internal quality assessment was performed calculating the prevalence of responses to an ad-hoc questionnaire measuring the level of satisfaction with the activity, according to the five criteria of the activity design: motivation, social connectedness, methods and techniques used, emotional well-being, and research perspective. The data analysed in the present study are available at the following open science repository: https://osf.io/meyxg (accessed on 2 November 2022).

## 3. Results

### 3.1. Descriptive Statistics

The descriptive statistics of the sample are presented in Table 1. In the ANOVA, no significant differences were found comparing the program participants vs. non-participants in the different indicators of well-being according to gender, family situation, field of knowledge, or type of doctorate (international/national). The profile of the Third Half participants was representative and comparable to a control group of UAB’s doctoral students (see Table 2). No significant differences were obtained in the *t*-test analyses when comparing the participants’ gender, cultural background, or field of knowledge. Participants showed significantly higher scores in the POMS’s depression subscale before starting the activity when compared to the non-participants (*F(1,17)* = 4.32; *p* = 0.039). It was observed a prevalence of 50% in generalised anxiety symptoms among the participants before starting the activity, similar to the 53% of non-participants (*χ^2^* = 0.90; *p* = 0.765). The prevalence of depression symptoms was also similar, showing a 45% prevalence of moderately severe-to-severe symptoms among participants and a 40% prevalence among non-participants. No differences were observed in symptoms of clinical anxiety, with a global prevalence reaching 53%. 

### 3.2. Differences between Pre-Test and Post-Test

The differences between pre- and post-test are presented in Table 3. Significant increases were obtained in vigour (*z* = 3.03; *p* = 0.003), friendship (*z* = 2.25; *p* = 0.024), and positive affect (*z* = 3.20; *p* = 0.001). Significant decreases in the mean scores of anger (*z* = 2.38; *p* = 0.017), fatigue (*z* = 2.25; *p* = 0.024), depression (*z* = 3.57; *p* = 0.000), negative affect (*z* = 2.22; *p* = 0.026), and state anxiety (*z* = 3.19; *p* = 0.001) were also observed. In Table 4, we present the frequencies and percentages of participants with clinical symptoms of depression and anxiety. The results showed that percentages of both anxiety and depression symptoms generally decreased after the intervention, but the differences were not statistically significant for either anxiety (*χ*^2^ = 2.00, *p* = 0.157) or depression (*χ*^2^ = 0.27, *p* = 0.600) symptoms.

### 3.3. Internal Quality Assessment of the Learning Process

The descriptive statistics for each item formulated for the evaluation of the internal quality of the program can be consulted in Table 5, with answers coded using a Likert scale rating from 0 to 5, with 5 as the maximum satisfaction rating. The mean values of the 20 items varied between 3.71 and 4.96, showing higher levels of satisfaction in all the criteria assessed regarding motivation, the reduction of social isolation, and the adequacy of the methods and techniques used or in the perceived impact on their emotional well-being. The highest levels of satisfaction were seen in those items related to the social support received, the outdoor approach and the feelings of respect and recognition experienced in the intervention.

## 4. Discussion

The main goal of the study was to assess the impact of the pilot implementation of the Third Half program on doctoral students and to study the current mental health situation of doctoral students. First, in line with previous descriptive studies in other European Universities in Belgium or in the Netherlands [5,12], the UAB data show that 53% of doctoral students present at least two symptoms of clinical anxiety, and 40% show moderate-to-severe symptoms of depression. Results also confirm the same tendency in a study analysing PhD students of 26 countries and 234 institutions in the world [13], in which 41% of graduate students scored as having moderate-to-severe anxiety and 39% scored as having moderate-to-severe depression. Accordingly, our results confirm the worsening mental health levels of ECRs and the so-called PhD crisis [3,7,8,9,10,13,14,15]. 

Second, and most important, since it was the main goal of the present pilot study, the Third Half intervention has shown encouraging results on its first pilot implementation and assessment. It is worth noting that the results have been satisfactory, not only in terms of the intervention impact, but also in terms of the positive experience of coordination between the different local teams involved and the satisfaction of the doctoral students who participated in this pilot experience. A repeated-measures analysis of variance has shown (after six sessions provided over 6 months) higher well-being and lower psychological distress scores among PhD students who participated in the Third Half. Specifically, significant increases were observed after the program in vigour, friendship, and positive affect, while significant reductions were obtained in anger, fatigue, depression, state anxiety, and negative affect. Also, symptoms of anxiety decreased, and even though statistical significance was not reached, reductions in depressive symptoms should also be highlighted. The reasons for this significant impact point to the multicomponent and evidence-based nature of the Third Half intervention: on the one hand, it is framed by the motivational model of self-determination [41] that is widely validated as a model for self-growth and for behavioural change [69,70] and by the humanistic approach to education [42,43,44,45,46] that guarantee social support and unconditional acceptance as key elements for a successful psychological intervention. On the other hand, the Third Half combines the inclusion of five key elements for psychological well-being that have widely and consistently demonstrated empirical evidence in the reduction of psychological distress and in the promotion of well-being: (a) Gamified activities [47,48,49,50] have added the integration of game elements to engage and motivate the process while having fun using traditional school-yard games, a factor that has probably contributed to an increase in ECR’s positive emotions. (b) Outdoor activities [52,53] have provided a more stimulating learning environment away from labs, classes and offices that might have facilitated the intervention impact through an exposure to natural environments and atmospheres that facilitate relaxation and anxiety reduction. This element has also added a sustainable use of different natural allocations by giving an additional therapeutic value to natural spaces of the campus. (c) Positive psychology applied in educational settings [54,55,56,57] has confirmed its impact combined with other techniques by using activities validated for the promotion of psychological well-being, such as counting blessings or identifying personal strengths. Although some studies have explored doctoral students’ perceptions of factors that promote their psychological well-being during the doctoral journey [69], to date no studies had previously been published analysing the impact of positive psychology among doctoral studies. However, more research should be performed to study the effectiveness of positive psychology alone to disentangle to what extent it contributes to well-being promotion and to contrast it with other approaches such as mentoring or coaching independently. Nevertheless, combining all of them in a multicomponent approach has proven to have a positive impact too. Coaching psychology and the Grow interviews [53,54,55], as one of the methods included in the motivational model of self-determination that frames the intervention, has allowed ECRs to focus on and monitor their personal goals, moving towards behavioural change and a healthy academic career, and aligns with previous pioneering results on the positive impact of coaching in doctoral studies in different European and non-European universities [71,72,73]. Coaching techniques are widely used and have shown its impact in other workplaces [74]; this impact is believed to be mediated by its association with the fact that coaching focuses on self-efficacy management, a cognitive mechanism that allows individuals to believe in their capacity to change and act in the necessary ways to reach their specific goals [70]. (d) The inclusion of low–moderate physical activity using gamified school yard games that promote behavioural activation has also added a component of motivation towards healthy behaviours, since it is well-known that physical activity is a protective factor for both mental and global physical health [59,60], and its practice is consistently recommended to enhance and maintain optimal health levels. Finally, (e) the inclusion of peer mentoring and peer support [61,62] by organising informal activities such as having a drink or a coffee after the intervention and by allowing participants to connect with each other, including trainers with similar experience, thus promoting informal social support, has allowed ECRs to share their worries and concerns, and foster bonds of friendship between them. Offering this activity as a time and space for reflection and informal dialogues might have also contributed to diminish ECRs’ feelings of social isolation, increase identification with others, friendship, altruism, and cooperation. 

Accordingly, this innovative pilot intervention has put together five evidence-based components for psychological improvement, and as hypothesized, has been effective in increasing ECRs’ well-being and decreasing psychological distress. It should be noted that the intervention axes of the Third Half had the fundamental objective of reducing social isolation and loneliness, being main risks for depression and anxiety among students during and after the pandemic [11]. It is also worth noting, in terms of quality assessment, that participants valued the intervention satisfactorily, with scores near the upper limit of the rank in the assessment of the learning process, as can be seen in Table 5. Considering that the pilot sample included both genders, different research fields and cultural backgrounds (it included a 40% of international PhDs), the Third Half seems to be effective for different researchers’ profiles, although further replications with wider samples are needed before reaching generalisations on its efficacy in other cultural backgrounds. The results motivate us to continue working together, with different local, national, and international teams, to overcome the so-called “PhD crisis” [3,7,8,9,10,13,14,15] and to keep on developing psychosocial programs that promote the well-being and mental health, not only of doctoral students, but of the entire research and academic community. National alliances such as the Catalan Net of Healthy Universities (including nine universities) [33] and international, such as the EuniWell (an alliance of eight European Universities) [75], the ARK-program (23 Norwegian and Swedish Universities) [76] or the ReMO project [1] are already leading the change by implementing strategic actions, research plans and policy imperatives towards building healthier research and academic environments [25]. These alliances are building a framework for the processual work with organizational mental health and well-being research in doctoral education, focusing and highlighting the need of preparation, screening, development of action plans, implementation, and evaluation of interventions. However, studies on the implementation and evaluation of interventions are still scarce, and pilot experiences such as the Third Half might contribute to an increase in data collection on implementations and assessments of interventions driven top-down from universities caring for their employees and willing to share best practices. Accordingly, the present pilot results open up a new window for sharing, implementing, and assessing multicomponent and evidence-based psychological interventions in public universities working towards more sustainable and healthier environments for researchers. Initiatives such as the Third Half program could be regarded as a cost-effective solution to offer institutional support to doctoral students and to facilitate their mental health and optimal development in the ERA institutions [2,20,23]. Such initiatives could not only reduce the incidence of mental health problems in the researchers’ community, but also the dropout prevalence in doctoral studies [21,22,23]. 

In the case of the present study, the voluntary efforts of local university teams allowed the implementation of the program; however, it is concluded that more institutional effort is needed, as well as investments in replicating, structuring, and standardizing actions as permanent and sustainable services that favour the optimal development of ECRs in their workplace. These training services are not yet widespread within the ERA, in part because their implementation depends mostly on grants and projects to perform these actions [1,2,24,25]. Even though some pioneering universities are responding to European policies at a local level, collective efforts must continue to be made to guarantee the implementation of the European Charter and the Code of Conduct for Researchers [37]. More generalised structural actions are still needed, and these pilot results call for research institutions to prioritize investments in the implementation and assessment of psychosocial training and services within doctoral programs. The goal is collective and systemically affects all of the ERA community, thus, it is important to continue working together (both researchers and institutions) to build healthier working and learning environments for the well-being and development of the entire academic community) [1,24]. 

However, although the present results are encouraging, this study has some significant methodological limitations given its pilot nature. First, the sample size was small, and limitations regarding the generalizability of the results are present. Hence, further replications to test its efficacy with a wider sample of doctoral students of different profiles and universities, including different trainers and cultural backgrounds, are encouraged. Second, the lack of a randomised control comparison group makes it hard to draw firm conclusions, thus, the Third Half should be replicated with more robust designs, including control groups, and tested to observe to what extent it might impact researchers’ well-being in the short and long term. Future studies could also include objective indicators of performance and economic and social indicators of well-being to analyse its impact in the community at a systemic level.

## 5. Conclusions

Our findings encourage the Third Half replication to keep on testing its impact, but also suggest and recommend to research institutions the implementation and assessment of this type of well-being program, since the results of this pilot experience have been satisfactory and effective in responding to the strategic goals and challenges of the ERA regarding well-being promotion and mental health management in doctoral studies [1,2,24,25]. Investments in well-being promotion among educational settings, as a public health strategy to support sanitary systems, would not only benefit the academic community, but it would also benefit public health services as a preventive strategy that could reduce the need for clinical assistance in mental health. A need that has dramatically increased during and after the COVID-19 pandemic around the world [7,8,9,10,11]. Accordingly, further research on effective mental health training is highly encouraged to assess and analyse their impact on doctoral students’ mental health and to continue advancing towards more sustainable and healthier research environments in doctoral communities, as a priority of the ERA [1,2,24,25,37], but also as a development goal and an urgent call within higher education and developed and developing societies working for peace and prosperity [27,28,38].

## Figures and Tables

**Table 1 ijerph-19-16905-t001:** Sociodemographic and scientific profile of the Third Half participants.

	*n*	%
Gender		
Man	6	33.3
Woman	12	66.7
Non-binary	0	0
Family status		
Single	15	83.3
Married or in a stable relationship	3	16.7
Field of knowledge		
Health Sciences	6	33.3
Life sciences	4	22.2
Experimental sciences	2	11.1
Social Sciences	2	11.1
Arts and Humanities	3	16.7
Engineering and Architecture	1	5.6
Type of doctorate		
Non-international	10	55.6
International	8	44.4

**Table 2 ijerph-19-16905-t002:** Descriptive statistics of mental health indicators of the Third Half participants compared with non-participants.

	Non-Participants (*n* = 140)		Participants (*n* = 18)			
	*M*	*SD*	*M*	*SD*	*F*	*p*
Age	30.94	6.79	33.24	7.82	2.93	0.088
POMS						
Anger	25.97	11.84	26.82	10.62	0.15	0.698
Fatigue	10.54	5.71	11.61	5.77	0.96	0.328
Vigour	8.56	4.61	8.05	4.16	0.34	0.560
Friendship	14.36	4.00	14.35	4.35	0.00	0.988
Tension	9.11	5.81	9.41	5.02	0.07	0.779
Depression	6.31	5.04	8.38	5.80	4.32	0.039
PANAS						
Positive Affect	29.98	8.20	29.09	7.99	0.32	0.569
Negative Affect	22.01	8.79	24.47	8.98	2.13	0.146
STAI						
State Anxiety	30.19	12.85	34.59	10.03	3.45	0.65
GAD-2_Anxiety	2.64	2.03	2.44	1.87	0.25	0.613
% With 2 symptoms	53%		50%			
PHQ-9_Depression	9.01	6.65	10.00	5.77	0.62	0.431
% Without symptoms	29%		15%			
% Moderate	31%		30%			
% Moderately severe to severe	40%		45%			

Notes: POMS = Profile of Mood States; PANAS = Positive and Negative Affect Scale; STAI = State-Trait Anxiety Inventory; GAD = Generalised Anxiety Disorder; PHQ = Patient Health Questionnaire.

**Table 3 ijerph-19-16905-t003:** Mean scores of mental health before and after the Third Half.

	*M*	*SD*	*Z*	P
POMS				
Anger				
PRE	8.61	6.31	2.38	0.017
POST	4.94	3.90		
Fatigue				
PRE	13.28	6.34	2.25	0.024
POST	8.44	6.20		
Vigour				
PRE	7.06	4.19	3.00	0.003
POST	12.61	4.11		
Friendship				
PRE	13.44	3.01	2.39	0.017
POST	15.44	2.12		
Tension				
PRE	10.61	5.68	1.89	0.059
POST	6.83	5.00		
Depression				
PRE	9.22	4.18	3.57	<0.001
POST	3.50	3.31		
PANAS				
Positive Affect				
PRE	25.78	7.55	3.20	0.001
POST	34.72	7.45		
Negative Affect				
PRE	26.61	9.61	2.22	0.026
POST	19.33	8.06		
STAI State Anxiety				
PRE	36.22	8.78	3.19	0.001
POST	21.22	12.40		
PHQ-9 Depression				
PRE	9.35	5.53	1.58	0.114
POST	6.16	3.85		
GAD-2 Anxiety				
PRE	3.11	1.56	3.03	0.002
POST	1.50	0.92		

Notes: POMS = Profile of Mood States; PANAS = Positive and Negative Affect Scale; STAI = State-Trait Anxiety Inventory; GAD = Generalised Anxiety Disorder; PHQ = Patient Health Questionnaire.

**Table 4 ijerph-19-16905-t004:** Frequencies and percentages of clinical symptoms of depression and anxiety.

	*N*	Pre	*N*	Post
Anxiety GAD-2				
Without symptoms	12	67%	18	100%
With symptoms	6	33%	0	0%
Depression PHQ-9				
Without symptoms	3	17%	6	33%
Moderate	6	33%	9	50%
Moderately severe to severe	9	50%	3	17%

**Table 5 ijerph-19-16905-t005:** Descriptive statistics of the items assessing internal quality.

	*M*	*SD*
Increased perspective	3.71	1.13
Increased motivation	3.86	1.16
Coping with career	3.86	1.16
Increased positive emotions	3.93	1.43
Stablished goals	3.93	0.99
Increased my well-being	4.07	1.07
Increased emotional well-being	4.21	0.80
Enjoyed gamified approach	4.21	1.31
Increased social connectedness	4.29	1.06
Felt more motivated	4.29	1.06
Decreased isolation	4.36	1.27
Increased my self-knowledge	4.36	1.00
Feeling part of research community	4.36	0.92
Social support	4.64	0.74
Enjoyed group approach	4.64	0.63
Felt respected and valued	4.64	0.74
Good coach	4.64	0.63
Would recommend the activity	4.79	0.42
Enjoyed outdoor approach	4.86	0.53
Enjoyed forest bathing	4.96	1.79

## Data Availability

The data analyzed in this article is available at the open science repository: https://osf.io/meyxg (accessed on 2 November 2022).

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
