# Peer review of "The Third Half: A Pilot Study Using Evidence-Based Psychological Strategies to Promote Well-Being among Doctoral Students"

_ijerph, 2022, doi:10.3390/ijerph192416905_

Round 1
Reviewer 1 Report
The manuscript discusses the strategy to promote well-being life among doctoral students. From the higher education institution's interest, the aim of the study is interesting and important to some extent. The authors offer several programs participated by 25 doctoral students and found significant increases in emotional indicators and a decrease in psychological distress. The results are interesting and sound. The manuscript is well-written and deserves publication. However, it seems there is a fundamental question related to the number of student respondents which is only 25 persons. Is the results valid for all doctoral students and can be implemented for various doctoral students with diverse backgrounds, e.g., culture, nationality, etc. The authors should address this question with additional arguments by performing new further studies or by comparing other similar studies reported in the literature. After addressing this question, the manuscript can be recommended for publication.
Author Response
Dear reviewer 1:
Thank you for your comments and positive feedback. We are aware that the small sample size makes it hard to draw generalizations and firm conclusions, but as noted in both the title and the type of manuscript, this is a pilot study by which the institution first wanted to check its efficacy before implementing it in a more systemic and generalized level in future academic years. We are not sure yet about the timings of forthcoming implementations but we know that this study will be replicated, not just in UAB but also in other European and non-European Universities with diverse cultural backgrounds, as the first author is working with the Researchers’ Mental Health Observatory (ReMO-Working group 3) and several researchers have expressed their willingness to replicate this pilot study in other research institutions.
We have addressed this issue in the discussion, emphasizing that this manuscript is a pilot experience (both in the implementation but also in the quantitative assessment of its impact using valid and reliable psychometric tools) and the need of performing further cross-cultural studies to generalize the preliminary results. We have also included additional arguments regarding the need to implement these types of programs by comparing the design and pilot results with other similar studies. Please see changes in Discussion pages 11-13 highlighted in yellow.
Reviewer 2 Report
Dear Authors I read yur manuscript with intrest. I found it intresting and relevant. I advice some minor improvment befor publication.
is it possible shorten that introduction? please hilight the resrach gap and hilight your finding. Please add more detailes on sampling aprt and finally add resrach limitation.
good luck
Author Response
Dear reviewer 2:
Thank you for the suggestions. We have shortened the introduction (20 lines) but we had to introduce some other lines according to other reviewers’ suggestions. We have highlighted the research gap (please see page 3 highlighted in yellow), have added more details on the sampling part (page 5) and have improved research limitations (page 13).
Reviewer 3 Report
if the authors could expand the Discussion section and provide more detailed comments regarding the study results?
Author Response
Dear reviewer 3:
The discussion section has been improved and implications of the study results have been added (please see page 11-12 highlighted in yellow).
Reviewer 4 Report
The document talks about an important issue, but it needs to be reviewed.
Title: the text does not fix exactly with the title. Some reasons make us to say that, for example: (a) document doesn’t talk deeply about psychological strategies (introduction just gives information about the context and in “material and methods” we find a description, but there is not a theoretical reflection and the results are neither explained taking into account each of the psychological interventions); (b) although doctoral students are study subjects, text seems to talk about their condition as member of teaching staff (not exactly as researchers), so perhaps it would be more adequate to talk about “contracted doctoral students” or “doctoral students as active member of a research group”; (c) reading the article, perhaps it should be adequate to incorporate the term “Covid-19”; (d) the idea of “pilot” only is remembered at the end (419), it has not be taken into account in the text.
We suggest to review title and, specially, text in order to get more cohesion. And, in order to help authors to improve the text, we write this text.
Introduction:
Text highlight Covid-19 influence, but Covid-19 it is not mentioned in abstract (neither in title). It should be useful to clarify if authors are worried about the direct effect of Covid-19 or if they are analyzing a problem that began before. In point 2.1. (144), it seems that The Third Half is in response to the impact of the pandemic.
Authors say that the risk among doctoral students is higher than the general population (14) and than the general highly educated adult population (44). Nevertheless, the introduction talks about the effect of the university policy and labor system (law salaries, job insecurity, …). Hence, the problem should be limited in the area of academic personal and it is not introduced from psychosocial perspective. Some questions that seem could be treated are: Does contracted doctoral students/active doctoral students work in harder conditions than academic established personal? Or what factors harm the mental health of doctoral students?
The structure of the introduction can be improved. It seems to have two final paragraphs: first one (52-56) and second one (82-85)
Roll of the authors:
Are authors members of UCAA? In line 138, we understand that they are the responsible of The Third Half, but in line 149 we can read “The UCAA agreed and designed a program called the Third Half”. In line 260, document talks about a collaboration with the UCAA. So the question is who designed the empirical work? In line 224, text talks about a document and it seems that authors participated in the pilot project, but it is not clear in the document. This relationship (authors-UCAA and pilot project) should be clearer.
Document talks about “trainers”: they were “PhD psychologists” and a “doctoral student, psychologist specialized in gamification” was also selected to support. Hence, in which way authors did participate in that process? Did authors participate as trainers? How they explain the project to trainers?
Sample and results:
The size of the sample is a problem. Authors talk about methodological limitations (473) mentioning “sample size”. Perhaps it would be advisable to have more applications of the program to write a scientific article that improve knowledge? Perhaps it is necessary more studies about the internal quality of the program.
Sometimes document does not seem the result of a research. Instead, it seems the description of a pilot program applicated in UAB and the opinion of the 18 participating students (analyze of the level of satisfaction after training).
The sociodemographic characteristics has not been taking into account to understand, for example, if there are differences from gender perspective.
It is recommended to assess whether to talk about a sample formed by 18 people it is necessary to use percentages.
Discussion and conclusion:
The content of the final discussion is very limited in terms of the scientific progress that allows. Conclusions are redundant. Low quality.
Other questions:
References are mentioned but, in most cases, article does not give information about the idea authors have selected of them (readers have to read the document mentioned to know it).
In general terms, we feel a lack of a critical view of the work itself (taking more into account psychology discipline), which would increase its quality.
Author Response
First of all, we want to thank reviewer 4 for all thorough suggestions raised. Please, find below all the answers addressing the issues commented.
Title: the text does not fix exactly with the title. Some reasons make us to say that, for example: (a) document doesn’t talk deeply about psychological strategies (introduction just gives information about the context and in “material and methods” we find a description, but there is not a theoretical reflection and the results are neither explained taking into account each of the psychological interventions); (b) although doctoral students are study subjects, text seems to talk about their condition as member of teaching staff (not exactly as researchers), so perhaps it would be more adequate to talk about “contracted doctoral students” or “doctoral students as active member of a research group”; (c) reading the article, perhaps it should be adequate to incorporate the term “Covid-19”; (d) the idea of “pilot” only is remembered at the end (419), it has not be taken into account in the text.
We suggest to review title and, specially, text in order to get more cohesion.
Answer: Please see the added comments along the text referring to the pilot study to fit better with the title and with the implications of it. (a) Please see highlighted in page 3 and in the discussion the linked between the title and the text (b) no, not necessarily the PhD students are part of the teaching staff, they are part of research staff, some of them are not even contracted. It has been added in the participants’ description as “doctoral students as active members of research groups”, (d) although it was performed during the Covid-19, it was not just in response to it, but rather to the general situation, (d) it has been added in different parts throughout the introduction and specially in the discussion.
And, in order to help authors to improve the text, we write this text.
Introduction:
Text highlight Covid-19 influence, but Covid-19 it is not mentioned in abstract (neither in title). It should be useful to clarify if authors are worried about the direct effect of Covid-19 or if they are analyzing a problem that began before. In point 2.1. (144), it seems that The Third Half is in response to the impact of the pandemic.
Answer: Indeed, the Covid-19 increased the problematic but it began before, and the Third Half did not just respond to the pandemic, but to the previous “PhD crises” situation. Please see some added sentences to clarify this in the introduction and discussion
Authors say that the risk among doctoral students is higher than the general population (14) and than the general highly educated adult population (44). Nevertheless, the introduction talks about the effect of the university policy and labor system (law salaries, job insecurity, …). Hence, the problem should be limited in the area of academic personal and it is not introduced from psychosocial perspective. Some questions that seem could be treated are: Does contracted doctoral students/active doctoral students work in harder conditions than academic established personal? Or what factors harm the mental health of doctoral students?
Answer: Thank you for these observations. As previous studies suggested, one of the reasons related to the decreasing doctoral students’ mental health is the labor system factors, such as working/academic conditions (instability and uncertainty, temporary contracts, workload, supervisor relationship, etc.) as psychosocial issues, that were already commented in the introduction (pages 1-2) as possible reasons. Please see its inclusion as psychosocial factors in e.g., Upton, J. (2013). Psychosocial Factors. In: Gellman, M.D., Turner, J.R. (eds) Encyclopedia of Behavioral Medicine. Springer, New York, NY. https://doi.org/10.1007/978-1-4419-1005-9_422 ).
It seems that doctoral students’ characteristics are not well expressed in text, so we have addressed this issue responding to the reviewer’s questions, please see it highlighted in yellow in the method-participants section (page 5).
The structure of the introduction can be improved. It seems to have two final paragraphs: first one (52-56) and second one (82-85)
Answer: The structure has been improved, please see it in text.
Roll of the authors:
Are authors members of UCAA? In line 138, we understand that they are the responsible of The Third Half, but in line 149 we can read “The UCAA agreed and designed a program called the Third Half”. In line 260, document talks about a collaboration with the UCAA. So the question is who designed the empirical work? In line 224, text talks about a document and it seems that authors participated in the pilot project, but it is not clear in the document. This relationship (authors-UCAA and pilot project) should be clearer.
Answer: Yes, some of the authors (2 first co-authors) are members of the UCAA, but not the rest of them. Please see this issue addressed in page 4 (program design). To note, the first author is the main author, who coordinated all the study and the team who implemented the program, assessment and publication of the results.
Document talks about “trainers”: they were “PhD psychologists” and a “doctoral student, psychologist specialized in gamification” was also selected to support. Hence, in which way authors did participate in that process? Did authors participate as trainers? How they explain the project to trainers?
Answer: As commented above, the two first authors were the trainers of the UCAA who both designed and implemented the training, and also co-wrote the final manuscript, with the rest of the co-authors, who participated in the design, analyses and writing of the manuscript. We have addressed this issue including a sentence for a better understanding of the trainers’ role in page 4 (materials and methods).
Sample and results:
The size of the sample is a problem. Authors talk about methodological limitations (473) mentioning “sample size”. Perhaps it would be advisable to have more applications of the program to write a scientific article that improve knowledge? Perhaps it is necessary more studies about the internal quality of the program.
Answer: As commented to the editor, we are aware that the small sample size makes it hard to draw generalizations and firm conclusions, but as noted in both the title and in the type of manuscript, it is a pilot study by which the institution first wanted to check its efficacy before implementing it in a more systemic and generalized level. We are not sure yet about the timings of forthcoming implementations but we know that this study will be replicated, not just in UAB but also in other European and non-European Universities with diverse cultural backgrounds, as the first author is working with the Researchers’ Mental Health Observatory (Working group 3) and several researchers have expressed their willingness to replicate this pilot study.
We also believe that it is important to acknowledge the nature of a pilot study. These are considered small scale preliminary studies conducted before any large-scale quantitative research publication. Examples of recent pilot studies including small samples can be the following: Flink et al., 2015, doi: 10.1016/j.sjpain.2015.01.005; Muro et al., 2022, doi: 10.1080/13416979.2021.1996516; Navarro et al., 2019, doi: 10.3389/fpsyg.2019.00055. Hence, the limited sample of the present manuscript (specified throughout the text) could be somehow expected.
We have addressed this issue in the discussion, emphasizing the pilot nature of the study and the need of replicating these first results in further studies and by comparing it with other published studies. Please see changes in Discussion pages 11-12 highlighted in yellow
Sometimes document does not seem the result of a research. Instead, it seems the description of a pilot program applicated in UAB and the opinion of the 18 participating students (analyze of the level of satisfaction after training).
Answer: The university policy makers wanted to implement well-being policies among doctoral studies, but first, the implementation and design, as well as its efficacy has to be assessed. It was a psychological training conceived as a pilot study that had to be tested before being definitely implemented at the UAB and probably to other universities in a near future that have already expressed their willingness to replicate it. For this reason, we believe that this pilot study could be already published as a pioneering experience that will help other researchers implement and assess it in their institutions. If no positive results had been obtained, the program would have not been considered for its further definitive implementation for coming years. As the reviewer can read in the methods and results section, before the opinion of 18 participating students in the final quality assessment of the learning process, the impact in their emotional well-being was also assessed using valid and reliable psychometric tools, so it is a pre-post pilot study analyzing psychological indicators with an additional quality assessment at the end of the process.
As commented to the editor when we were invited to publish in the IJERPH Special Issue, the document was first aimed to be a research-policy report, as that, policy-papers are rather oriented to stakeholders and policy makers, since well-being programs are still far to be implemented in doctoral studies and more evidence is needed to persuade policy-makers of research institutions to implement them. Accordingly, we need to gather and publish data on the efficacy of well-being programs to gain evidence and persuade policy-makers to keep on investing in these kind of interventions. Furthermore, policy-papers must highlight the reasons for recommending one policy (or set of policies) over others, using indicators and outcomes that support these recommendations.
The sociodemographic characteristics has not been taking into account to understand, for example, if there are differences from gender perspective. It is recommended to assess whether to talk about a sample formed by 18 people it is necessary to use percentages.
Answer: We performed previous t-tests to compare gender, cultural backgrounds (local vs. international) and field of knowledge, but no differences were found in the baseline levels of different mental health indicators. We have added this information in the results section. Regarding percentages, we had to standardize numeric results, so that further studies can compare the samples and findings.
Discussion and conclusion:
The content of the final discussion is very limited in terms of the scientific progress that allows. Conclusions are redundant. Low quality.
Answer: Discussion and conclusions have been addressed, please see it highlighted in yellow in different parts of the discussion.
Other questions:
References are mentioned but, in most cases, article does not give information about the idea authors have selected of them (readers have to read the document mentioned to know it).
Answer: We agree with the reviewer comment and we have added some more detailed information of each reference in order to address this issue.
In general terms, we feel a lack of a critical view of the work itself (taking more into account psychology discipline), which would increase its quality.
Answer: As commented above, it was conceived as a policy paper, not as a proper psychological paper, to encourage research institutions and stakeholders to invest more in well-being strategies to improve the current situation in mental health among young research staff. Limitations of the present manuscript have been provided in more detail, see it highlighted in yellow. Also, some more considerations regarding the psychological discipline itself have been included.
Round 2
Reviewer 1 Report
The manuscript has been considerably revised following our comments/suggestions. Based on the revisions made, I am happy to recommend the publication of the manuscript.
Reviewer 4 Report
Congratulations for your resarch! Thanks for study that subject!